

# HiCEnterprise: identifying long range chromosomal contacts in Hi-C data

Hanna Kranas[1,2,*], Irina Tuszynska[1,*] and Bartek Wilczynski[1]

[1] Institute of Informatics, University of Warsaw, Warsaw, Poland
[2] Current Address: Institute for Research in Biomedicine (IRB Barcelona), The Barcelona Institute of Science and Technology, Barcelona, Spain
* These authors contributed equally to this work.

## ABSTRACT

**Motivation:** Computational analysis of chromosomal contact data is currently gaining popularity with the rapid advance in experimental techniques providing access to a growing body of data. An important problem in this area is the identification of long range contacts between distinct chromatin regions. Such loops were shown to exist at different scales, either mediating relatively short range interactions between enhancers and promoters or providing interactions between much larger, distant chromosome domains. A proper statistical analysis as well as availability to a wide research community are crucial in a tool for this task.

**Results:** We present HiCEnterprise, a first freely available software tool for identification of long range chromatin contacts not only between small regions, but also between chromosomal domains. It implements four different statistical tests for identification of significant contacts for user defined regions or domains as well as necessary functions for input, output and visualization of chromosome contacts.

**Availability:** The software and the corresponding documentation are available at: github.com/regulomics/HiCEnterprise.

**Supplementary information:** Supplemental data are available in the online version of the article and at the website regulomics.mimuw.edu.pl/wp/hicenterprise.

# INTRODUCTION

Chromosomes in eukaryotic cells are very complex assemblies of nucleic acids and proteins that function in a tightly packed 3D environment of the cell nucleus (*Sazer & Schiessel, 2018*). The packing of chromosomes is at the same time dynamic and visibly different between cells in the same population, yet its conformation is proven to be non-random to allow for efficient activation and repression of subsets of genes defined by the dynamic epigenetic state of the cell (*Spector, 2003*). Scientists have been interested in studying the rules governing the chromatin structure and its dynamics for a long time; however, we were mostly limited to theoretical studies based on relatively sparse imaging data until the development of the Hi-C technique (*Lieberman-Aiden et al., 2009*). The body of data from Hi-C experiments is quickly growing, allowing us to answer more questions related to chromosome structure and its relation to gene regulation. In particular, the question of identifying chromosomal contacts have been studied both on

Corresponding author
Bartek Wilczynski,
bartek@mimuw.edu.pl

the level of small regions (even down to 1 kb) that could represent enhancer-promoter interactions (*Won et al., 2016*), as well as on the level of larger ones, like TAD-to-TAD interactions (*Niskanen et al., 2017*). While the enhancer-promoter interactions are widely studied, the importance of larger domain-domain interactions are not quite as well studied, however there are reports of important examples of such interactions having a regulatory function such as the polycomb related strong interaction in Drosophila (*Sexton et al., 2012*) or the cell-type-specific silencing interaction on human chromosome 17 (*Niskanen et al., 2017*). Even though a number of methods for chromatin contact detection were proposed, such as HOMER (*Heinz et al., 2010*), HiCCUPS(*Rao et al., 2014*), GotHIC (*Mifsud et al., 2015*) or Fit-Hi-C (*Ay, Bailey & Noble, 2014*; *Kaul, Bhattacharyya & Ay, 2020*) (see *Forcato et al. (2017)* for a review), there are still no freely available software tools for identification of those interactions between the larger segments, that is, domains (the only available method, PSYCHIC (*Ron et al., 2017*) requires Matlab and Statistics and Machine Learning Toolbox). Since more researchers are interested in identifying such contacts, it may be helpful for them to have a freely available implementation of statistical protocols that were adapted by us to identify long range chromatin interactions not only between points but also between chromosomal domains.

## METHODS

HiCEnterprise is a package consisting of two types of contact analyses: between short regions (1–3 bins of a Hi-C map) and between topologically associating domains (TADs) (*Pombo & Dillon, 2015*). Here regions are short genomic sequences, of length between 1 and 3 bins, where a bin represents the smallest DNA segment analyzed given the Hi-C map resolution. TAD is defined as a larger segment, within which the DNA sequences physically interact with each other more frequently than with sequences outside of it. TADs can be noticed on the Hi-C map as dark (contact-rich) triangles near the diagonal.

The first part of the package, `HiCEnterprise regions` is used for identification of statistically significant long range contacts between small regions. The analysis implemented in this part is based on the method for identification of bin-to-bin interactions and creating interaction profiles based on Hi-C data as introduced by *Won et al. (2016)*. As Hi-C cis-contact maps should be symmetrical, interaction profile for a region located in a particular bin is obtained by extracting intensities only horizontally, from the left and right of the region of interest positions on the diagonal. Significant contacts between bins are identified as enrichments under background distribution (fitted Weibull distribution matched by chromosome and distance, *Won et al. (2016)*). False Discovery Rate (FDR) calculated with a Benjamini–Hochberg procedure (*Benjamini & Hochberg, 1995*) is used to correct for multiple testing, giving us $q$-values in addition to the $p$-values of every potential interaction.

It still may be difficult to distinguish between true interactions and noise, so to provide additional test for the validity of found interactions, we offer an option for simultaneous analysis of multiple biological replicates. If several Hi-C maps (replicates) are provided on input, interactions are considered significant only if their FDR value is below the selected threshold in all replicates. In the example interaction profile plot (Figs. 1A, 1C, 1D)

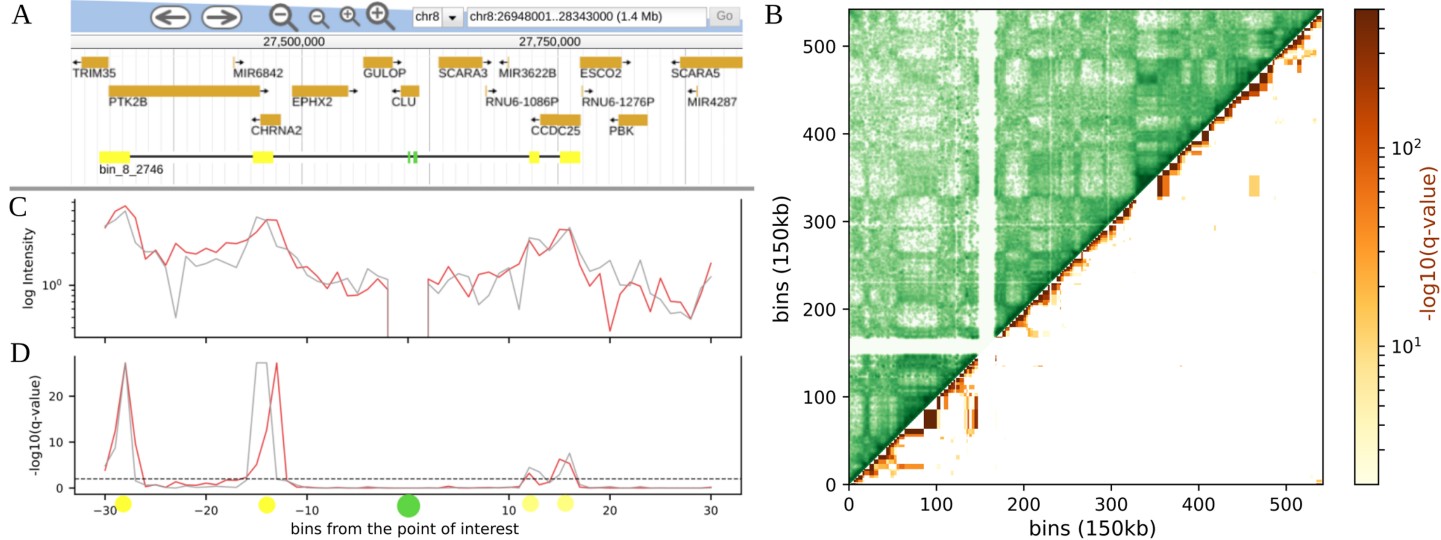

**Figure 1** Example of visualization of long range contacts for: (A) regions, here enhancer-promoter contacts, for two 10 kb resolution Hi-C maps from human Fetal Brain cells. Contacts were calculated up to 30 bins distance each way from the point of interest. Top to bottom: a Jbrowse (*Skinner et al., 2009*) screenshot with genes and interaction profile representation (points of interest in green, their contact predictions in yellow); HiCEnterprise interaction profile plot with intensities (weighted by distance) and −log10 of FDR corrected *p*-values (*q*-values) with a threshold set at 0.01. (B) TADs, here HiCEnterprise visualization for 150 kb resolution Hi-C map of 17th chromosome from HUVEC. Left upper trianle: original Hi-C map contact frequencies; right lower triangle: −log10 of *q*-values of the interdomain interactions calculated with hypergeometric distribution.

one can see the analysis for two Fetal Brain maps (*Won et al., 2016*), where for two enhancers located within one 10 kb bin, four significant interactions with potential genes of interest have been found, confirmed by two replicate maps. The results shown are consistent with the results from the original article (*Won et al., 2016*).

To run this point interactions analysis, it is required that the user provides at least one Hi-C chromosome map in the numpy format and a BED file with coordinates of regions to extract interaction profiles. Since enhancers are distal regulatory elements that affect transcription levels of target genes by being in physical close proximity to them, an example use of the HiCEnterprise region mode would be to find DNA regions frequently in contact with particular enhancers based on Hi-C maps. As this mode will be likely often used exactly for the purpose of scanning for enhancer related interactions, we have provided also a mode in which the user can supply as input a list of enhancers from a FASTA file obtained from EnhancerAtlas (*Gao et al., 2016*), a database that provides annotation of enhancers in the human genome. HiCEnterprise can plot the results related with found interactions either with matplotlib (*Hunter, 2007*) or rpy2 (*Gautier, 2012*). Output files with found interacting regions are available in three formats: txt, BED-like and GFF-like. Remapping between assemblies is possible for BED and GFF files (pyliftover package is required: https://pypi.org/project/pyliftover/).

The second part of HiCEnterprise, `HiCEnterprise domains` is dedicated to calculation of long range interactions significance scores for pairs of larger chromosome segments, like the TADs in Hi-C contact maps, as introduced in (*Niskanen et al., 2017*). In this mode the user needs to define borders of TADs obtained using external

software (e.g., HOMER by *Heinz et al. (2010)*; *Dali & Blanchette (2017)* for more examples). As the first step of this method, the new matrix M (with the shape of $N \times N$ where $N$ is the number of domains in the chromosome considered) is calculated. M[$i,j$] represents the total number of Hi-C contacts for the pair of domains $i$ and $j$. Next, for each pair of domains in the new matrix, a *p*-value is calculated based on the hypergeometric, Poisson or negative binomial test.

Our software calculates the parameters of the chosen distribution based on the data observed in the actual Hi-C matrix and calculates a *p*-value for enrichment under the null model. Similarly to the point analysis, *p*-values are converted to FDR *q*-values to account for multiple hypothesis testing. HiCEnterprise domains mode, that calculates the probability of inter domain contacts based on Hi-C maps, can be run with different options. Some of them are necessary to run the program: the chromosome number, the Hi-C map, the resolution of Hi-C map, the file with domain borders information and the domain level if a hierarchical TAD caller like Sherpa (*Krolak & Wilczynski, 2012*) was used to determine the domain borders. It is possible to modify the threshold of the *q*-values that will be considered as significant and returned in the output file (the default value is 0.01). By default, the resulting *p*- and *q*-values are written to two text files. The user can also choose to make a plot with results as a contact map, with significant contacts between domains highlighted with color (as seen in Fig. 1B). One can also change colors of the contact maps and/or interactions on the figure, propose the title or change the distance between ticks on the generated figures.

## RESULTS

Unlike the native structure from crystallographic studies in the case of proteins or small nucleic acid molecules, chromatin has no native structure (*Hajjoul et al., 2013*). This makes the task of identifying domain contacts more subjective. So far, there is no gold standard to compare methods that predict the interactions of chromatin to, so it is only possible to compare available methods between each other. We compared the results of HiCEnterprise regions mode, HiCCUPS (*Rao et al., 2014*) and Homer (*Heinz et al., 2010*) between each other and did a short quantitative analysis.

For this task, we used human umbilical vein endothelial cells (HUVEC) Hi-C maps with 25 kb resolution, that were kindly provided by Henri Niskanen (described in our joint publication *Niskanen et al. (2017)*). We compared the interactions found by the aforementioned tools within +/− 100 bins around the region of interest for human chromosome 1. First, we found interactions by HiCCUPS and HOMER: HiCCUPS found interactions for 262 regions while Homer for 204 regions. Next, we combined HiCCUPS and Homer results and removed the duplicate regions to prepare an input file with regions of interest for HiCEnterprise regions mode. Our tool found interactions for 386 out of 396 given regions. Some regions of interest are indicated as interacting with more than one site, so we compared all possible interactions (1,446 for HiCEnterprise, 283 for HiCCUPS and 209 for Homer) and found out that HiCEnterprise recognized almost all contacts that were found by HiCCUPS and Homer programs, and proposed many additional statistically significant interaction places (Fig. 2A). Summing up, for 252 regions

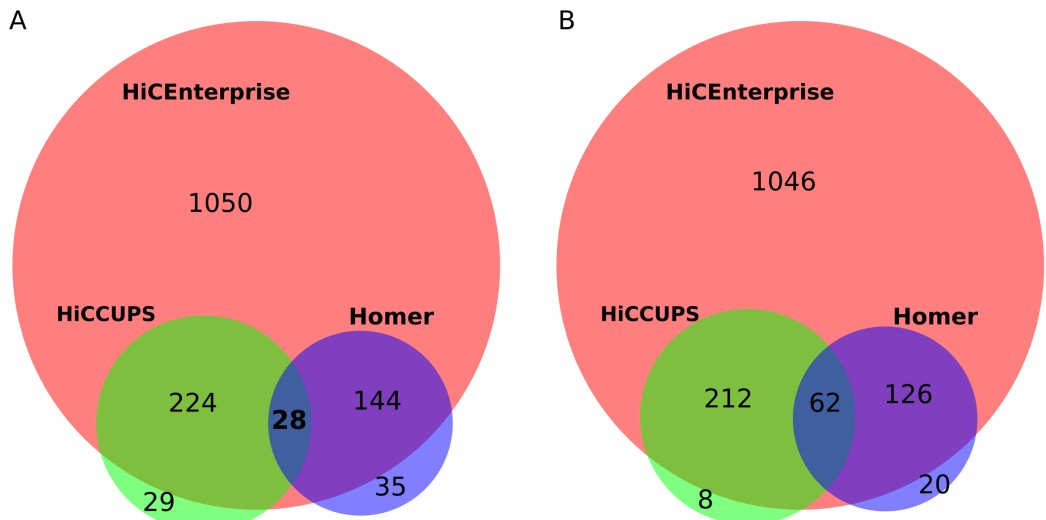

**Figure 2** (A) Venn plot for comparing results intersection of HiCEnterprise with HiCCUPS and Homer programs. (B) Venn plot for comparing results intersection of HiCEnterprise with HiCCUPS and Homer programs with one bin freedom (if programs found the same bin +/−1, the contact is considered as common).

HiCEnterprise and HiCCUPS found the same interaction bin on the Hi-C map ($p$-value using binomial test = $3.17 * 10^{-295}$), while HiCEnterprise and Homer found the same bins of HiC map for 172 regions with $p$-value $1.21 * 10^{-171}$, whereas HiCCUPS and Homer found 28 identical interaction sites with $p$-value $7.49 * 10^{-30}$.

Sometimes, two methods found interactions between neighboring bins, differing just by 1 position on the chromosome, which may be explained as identification of a similar interaction, given that the enrichments are frequently spanning more than one bin. We checked how often this is the case, treating predicted interactions from different tools that fell within the +/− 1 bin from each other as identical. As Homer had the smallest number of predicted interactions, we compared its results with the other two tools, and then checked common interactions for HiCCUPS and HiCEnterprise programs. With this approach, the commonalities between the three considered programs are even more obvious (Fig. 2B): we have 62 common interactions instead of 28, and there are even fewer individual interactions found either only by HiCCUPS (8 instead of 29) or by Homer (20 instead of 35).

As HiCEnterprise regions mode found many more significant interactions than the other tools we tested, we wanted to see how confident we can be in the functionality of those predictions. Thus, we overlapped the found interacting points from the aforementioned analysis (not the points of interest) with CTCF and RAD21 (cohesin component) Chip-seq peaks from HUVEC cells, provided by *Niskanen et al. (2017)*. We discovered that those interactions are significantly enriched over all queried bins in CTCF (chromatin binding factor) and RAD21 (cohesin complex component protein) Chip-seq peaks ($p$-value using binomial test = $1.16 * 10^{-71}$, $8.27 * 10^{-75}$ accordingly). Additionally, we looked at how many of the predicted HiCEnterprise region-wise interactions are forming longer stretches of consecutive predictions. For the total of

1,446 unique origin-interaction bins found, after clustering, we obtain 849 clusters of consecutive interactions. While most of those clustered interactions are constrained to up to 3 bins (probably due to the interacting region lying somewhere in-between those bins), we also find few very long interactions, up to 9 bins. We think that those might be reflection of larger, domain-wise interactions, however, thanks to the statistical model we employed here, the larger domain-wise interactions are not affecting the region-wise results in a significant manner.

To the best of our knowledge, there is no other freely available method for prediction of TAD-TAD interactions based on Hi-C maps that we could compare to. However, we wanted to understand if domains from one chromosomal compartment interact with domains in the same compartment more often than expected by chance. Chromatin compartments A and B, are usually defined by the first principal component (PC1) from Principal Component Analysis (PCA)—PC1 positive and negative values describes the A (active and accessible) and B (passive and closed) chromatin compartments accordingly (*Lieberman-Aiden et al., 2009*).

We started this analysis by detecting domain borders using Sherpa algorithm (https://github.com/regulomics/sherpa) developed by our group for the same 25 kb HUVEC Hi-C map of chromosome 1. Next, we calculated the probability of interactions between domain pairs using HiCEnterprise domain mode, using the three available distributions. We used PCA to identify the compartments to which the interacting domains belong. Each domain has been classified as belonging to a positive or negative compartment based on the number of positive or negative PC1 values within the domain. For all three methods, chi-square test rejected the null hypothesis according to which observed values would have the expected frequencies ($\frac{n^2}{2}$ for negative–negative pairs and $\frac{p^2}{2}$ for for positive–positive, and $p * n$ for positive–negative, where $n$ i the number of negative domains, and $p$ of positive). Next, binomial tests have shown the depletion of positive-negative pairs to be statistically significant ($p$-values $\leq 3.14 * 10^{-26}$, Fig. S3)— which is in line with our expectation, as it has been shown that domains from one compartment tend to interact with domains from the same compartment.

Detailed description of all the analyses and results can be found in the Supplemental File HiCEnterprise_VS_HiCCUPS_Homer.pdf.

## DISCUSSION

The software presented here is a flexible tool for identification of interacting loci based on Hi-C experiments, and the first freely available for calling domain-domain interactions. We provide two different functionalities (region-to-region and domain-to-domain contact identification) with several statistical tests that were already shown to be appropriate for each of the scenarios (Weibull distribution for regions and hypergeometric, Poisson and negative-binomial tests for domains analysis). This is in contrast to some earlier approaches like Fit-Hi-C that fit splines to the data or HiCCUPS that simply search for locally enriched regions in Hi-C maps.

HiCEnterprise regions mode is unique compared to other available tools—it finds statistically important interactions (pixels on the Hi-C map) between regions compared

with other interactions at the same distance (in one diagonal of Hi-C map), HiCCUPS analyses the contacts between the regions of chromatin against the number of contacts in a series of regions surrounding the pixel, while Homer script identifies chromatin interactions by comparing local maxima with both local surrounding region and global interactions for the same genomic distance. Therefore it is in line with our expectation that programs that use different statistical models to calculate the interaction between regions on Hi-C maps not always found the same contact bins. However, in most cases HiCEnterprise found the same interactions as both HiCCUPS and Homer programs (Fig. 2A), while there are also regions, where programs found different contacts. Moreover, these point interactions found by HiCEnterprise are significantly enriched in CTCF and RAD21 Chip-seq peaks further suggesting their validity. Lastly, although there is no competing tool for the HiCEnterprise domain mode that we know of so far, we managed to show that the significantly interacting domain pairs are more often than expected belonging to the same chromosomal compartment.

In summary, the strength of our approach for both modes comes with assigning $p$- and $q$-values to Hi-C interactions, allowing the user to generate their own hypotheses, combined with additional functionalities that might help give additional evidence for the found interactions. By providing a tested and easy to use implementation, we hope to make it easier for experimentalists to use these methods without the need to implement them on their own.

## ACKNOWLEDGEMENTS

We would like to thank Minna Kaikkonen and Henri Niskanen from the University of East Finland for providing us with Hi-C and ChIP-Seq data for testing purposes.

### Funding

This work has been supported by the Polish National Science Center Grant decision number (DEC 2015/16/W/NZ2/00314). The funders had no role in study design, data collection and analysis, decision to publish, or preparation of the manuscript.

### Grant Disclosures

The following grant information was disclosed by the authors:
Polish National Science Center: DEC 2015/16/W/NZ2/00314.

### Competing Interests

The authors declare that they have no competing interests.

### Author Contributions

- Hania Kranas conceived and designed the experiments, performed the experiments, analyzed the data, prepared figures and/or tables, authored or reviewed drafts of the paper, and approved the final draft.

- Irina Tuszynska conceived and designed the experiments, performed the experiments, analyzed the data, prepared figures and/or tables, authored or reviewed drafts of the paper, and approved the final draft.
- Bartek Wilczynski conceived and designed the experiments, analyzed the data, authored or reviewed drafts of the paper, and approved the final draft.

## Data Availability

The software and its documentation is available at GitHub: https://github.com/regulomics/HiCEnterprise.

## Supplemental Information

Supplemental information for this article can be found online at http://dx.doi.org/10.7717/peerj.10558#supplemental-information.

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
