# Peer review of "HiCEnterprise: identifying long range chromosomal contacts in Hi-C data"

_PeerJ, doi:10.7717/peerj.10558_

## Round 0.1 · original submission · Minor Revisions

The manuscript provides some novel elements for analyses around the identification of long-range chromosomal contacts from Hi-C data. However, it is not the first software to offer the ability to identify long-range chromosomal contacts. In addressing the suggestions and concerns of the reviewers, I suggest that particular attention be paid to the following:

1) Acknowledging the existing software packages available for this type of work, and providing clear comparison and contrast with these software and the proposed method here.

2) Providing a brief description of the data set used as an example - this information could be pertinent to future users in assessing the applicability or efficacy of this method.

3) Adding more to the results section - a table or more detailed summary of the findings would be of benefit to readers, rather than assuming they will look for the supplementary data.

4) More detail regarding analyses of domains - why is this useful?

·

Basic reporting

Im not sure that this is the first freely available tool for long range chromosome contacts between domains, is that true? I think this is capable in juicebox, which is also free. What should be advertised is that this is a step above current free software by injecting statistical vigor to what is currently available. The authors provide a novel tool to attach P and Q values to HiC interactions, something that I have not seen. The software has a long list of software dependencies, but it is provided through a python installer

Figure 1b needs a label for the distance illustrated in the y and x axis.
Line 67-73 The enhancer atlas needs defined more thoroughly, (what is it?, usefulness?) This paragraph needs one or two more sentences to be clearer.
Figure S1 needs to have a label for the axes. How does it display p-value?

Line 46 The distinction between region and domain is not clear.
Line 68: Most will not know numpy format, but many will know binary.
Line 85-93 Domain level is not clear to the general reader.
Line 95: “Chromatin has no native structure” seems speculative and needs references.
Line 102: HUVEC needs to be defined.
This line in the supplemental “proposed much more statistically
important and hypothetically interesting contacts” seems speculative, how are they more important?

Experimental design

no comment

Validity of the findings

no comment

Additional comments

While the supplemental tutorial is nice, it is more to understand how the software works, rather than a tutorial. The best source for tutorials is their online presence, both tutorials, and a github repository to display their code.
Figure S2 could also be a misassembled inversion, or an inversion between the genome accession and the HiC DNA-seq.
The HiC-DC software errors are likely due to a version error in R or in an R package.

Reviewer 2 ·

Basic reporting

Please provide some form of graphical summary of the results of your cross-comparison with Homer and HiCCUPS, ideally by plotting the results of each tool as you do in figure 1.

Fig 1: Please add units to axes of Fig1B. Is it 0 to ~550kbp? Please also add some details of the genome and location within genome of the data you display in the figure caption.

Experimental design

I feel the manuscript could use a more detailed discussion of the exact problem HiCEnterprise is trying to solve. In particular, an explanation of the two modes of operation (region vs domain), how they differ, and what biological problems each attempts to solve would be very useful.

Similarly, the discussion could use a brief overview of how the tool performs relative to others, and when a biologist should use this tool instead of others.


the example dataset is not discussed in sufficient detail. please provide a few sentences

While not required for publication, I suggest the authors submit the HiCEnterprise python package to pypi and bioconda to ease installation.

Validity of the findings

In multiple places the manuscript refers to HiCEnterprise being the only tool which can identify long-range contacts from HiC data. While I am not very familiar with this specific area of bioinformatics, a cursory literature search found at least one other tool which seems to be targeted at the same problems as HiCEnterprise (e.g. FitHiC2; Kaul et al. Nature Protocols 2020; doi: 10.1038/s41596-019-0273-0). Please update the manuscript to either describe that HiCEnterprise addresses a subtly different problem and therefore is unique, or perform experiments that compare HiCEnterprise to existing tools and show how HiCEnterprise outperforms these tools.

I would also be interested to see how HiCEnterprise, Homer, and HICCUPPS performed on data consistent with a null hypothesis of random interaction/"negative control". The most obvious approach to this would be to shuffle the coordinates of one end of the paired association such that "ends" of the hiC read data seem randomly associated, however the authors may have more intelligent and realistic ideas that explore the null distribution of HiC data.

Additional comments

There are many missing spaces in the abstract where it seems text was copy-pasted from a PDF, removing spaces between lines. please double check the text in the peerj website.

·

Basic reporting

The article is mostly well written with few typos and grammatical errors.

Literature is well referenced, although there is a few more tools available for annotation of long-range interactions (for example, cLoops doi: 10.1093/bioinformatics/btz651 and chromosight doi:10.1101/2020.03.08.981910).

The structure of the article slightly deviates from the PeerJ standard, however it is well justified by a purely computational nature of the presented work. Code and data used for this paper is available. However the Results section only references Supplementary figures, and in my opinion there should be a main figure with key findings.

The figures could benefit from more detailed legends, and annotations (e.g. text from lines 62-65 should be in the legend, not main text, and plots need to have their axes labelled, et cetera).

Experimental design

The purpose of this work was to implement previously described approaches to discovery of regions with significantly enriched interaction frequency in Hi-C data in one unified framework, and make them more accessible to use. The work appears to be performed well and the produced software will be useful for the field.

However the paper would benefit from more detailed explanation of how the analysis of interactions between extended domains could be useful. Does it identify anything different from finding compartments? Additionally, the article would benefit from more examples of identified interactions, and some analysis about them, e.g. their size distribution, or distance separations between interacting regions.

It would be great if the authors could implement input of Hi-C data based on more standard and flexible formats, in particular cooler from Mirny lab (https://github.com/mirnylab/cooler), or .hic from the Lieberman Aiden Lab.

Validity of the findings

No comment

---

## Round 0.2 · accepted · Accept

The revised submission addresses the reviewer concerns and details some elements of the reviewer suggestions that are out of scope for this manuscript. I believe the manuscript provides a timely tool for HiC analyses as they continue to increase in use.